# Changes in non-communicable diseases, diet and exercise in a rural Bangladesh setting before and after the first wave of COVID-19

Carina King[1]*, Sanjit Kumer Shaha[2], Joanna Morrison[3], Naveed Ahmed[2], Abdul Kuddus[2], Malini Pires[3], Tasmin Nahar[2], Raduan Hossin[2], Hassan Haghparast-Bidgoli[3], A. K. Azad Khan[2], Justine Davies[4], Kishwar Azad[2], Edward Fottrell[3]

1 Department of Global Public Health, Karolinska Institutet, Stockholm, Sweden, 2 Diabetic Association of Bangladesh, Dhaka, Bangladesh, 3 Institute for Global Health, University College London, London, United Kingdom, 4 Institute of Applied Health Research, University of Birmingham, Birmingham, United Kingdom

* carina.king@ki.se

## Abstract

Prevalence of non-communicable diseases (NCDs) is high in rural Bangladesh. Given the complex multi-directional relationships between NCDs, COVID-19 infections and control measures, exploring pandemic impacts in this context is important. We conducted two cross-sectional surveys of adults ≥30-years in rural Faridpur district, Bangladesh, in February to March 2020 (survey 1, pre-COVID-19), and January to March 2021 (survey 2, post-lockdown). A new random sample of participants was taken at each survey. Anthropometric measures included: blood pressure, weight, height, hip and waist circumference and fasting and 2-hour post-glucose load blood glucose. An interviewer-administered questionnaire included: socio-demographics; lifestyle and behavioural risk factors; care seeking; self-rated health, depression and anxiety assessments. Differences in NCDs, diet and exercise were compared between surveys using chi2 tests, logistic and linear regression; sub-group analyses by gender, age and socio-economic tertiles were conducted. We recruited 950 (72.0%) participants in survey 1 and 1392 (87.9%) in survey 2. The percentage of the population with hypertension increased significantly from 34.5% (95% CI: 30.7, 38.5) to 41.5% (95% CI: 38.2, 45.0; p-value = 0.011); the increase was more pronounced in men. Across all measures of self-reported health and mental health, there was a significant improvement between survey 1 and 2. For self-rated health, we observed a 10-point increase (71.3 vs 81.2, p-value = 0.005). Depression reduced from 15.3% (95% CI: 8.4, 26.1) to 6.0% (95% CI: 2.7, 12.6; p-value = 0.044) and generalised anxiety from 17.9% (95% CI: 11.3, 27.3) to 4.0% (95% CI: 2.0, 7.6; p-value<0.001). No changes in fasting blood glucose, diabetes status, BMI or abdominal obesity were observed. Our findings suggest both positive and negative health outcomes following COVID-19 lockdown in a rural Bangladeshi setting, with a concerning increase in hypertension. These findings need to be further contextualised, with prospective assessments of indirect effects on physical and mental health and care-seeking.

**Data Availability Statement:** The data used in this paper are in a fully anonymised format at: https://doi.org/10.6084/m9.figshare.19337831.v1.

**Funding:** The trial is funded by the UK Medical Research Council (ref: MR/T023562/1) under the Global Alliance for Chronic Diseases (GACD) Diabetes Programme - EF. The funders had no role in study design, data collection and analysis, decision to publish, or preparation of the manuscript.

**Competing interests:** The authors have declared that no competing interests exist.

## Introduction

Globally, the prevalence of type-2 diabetes mellitus (T2DM) has been increasing and is one of the biggest health challenges, particularly in Asia [1]. The morbidity burden has been exacerbated by the COVID-19 pandemic, with uncontrolled hyperglycaemia and T2DM as significant risks for severe COVID-19 disease and mortality [2–4]. It has also been hypothesised that SARS-CoV-2 infections could trigger diabetes onset [5], and post-infection syndromes (i.e. long-COVID) include depression and persistent multi-organ impairment [6, 7]. Other common non-communicable diseases (NCDs), including obesity, hypertension and heart disease are also risks for COVID-19 mortality [8–10].

Non-pharmaceutical interventions which restrict movement and close contacts to try and contain COVID-19 transmission however could worsen NCD risks and management. For example, studies from Bangladesh have reported that older adults with NCDs experienced barriers in accessing care [11], and people living with diabetes were concerned about increased risks and consequences of COVID-19 infections [12]. Negative impacts on mental health, physical activity, diet and income have been reported in a range of South Asian contexts [13, 14], and the link between poorer mental health and diabetes increases the vulnerability of this group [15]. Therefore, improving T2DM management and reducing NCD risks may be effective in lessening the impact of COVID-19 in these populations [3, 16].

Bangladesh reported its first confirmed case of SARS-CoV-2 on the 8[th] March 2020, and entered into a national government declared lockdown from the 23[rd] March. The lockdown involved the closure of non-essential workplaces, schools, and places of worship and advice was to stay at home. The lockdown ended on 30[th] May 2020 but was followed by restrictions on mass gatherings which ended in September 2020 [17]. By the 1[st] January 2021 there had been 514,500 confirmed COVID-19 cases, and 7576 registered deaths, with the epicentre in Dhaka and cases confirmed in all districts [18]. The second wave began in March 2021 and a new lockdown was implemented on 5[th] April 2021.

In rural Bangladesh, where more than one third of adults have raised blood glucose levels and almost half have a raised blood pressure [19], understanding the direct and indirect impacts of the COVID-19 pandemic on NCDs is important. We therefore aimed to compare the prevalence of common NCDs (T2DM, hypertension, obesity, depression and anxiety) and measures of self-rated health, diet and exercise, immediately before the COVID-19 outbreak began and one year later. These provide a snapshot of potential indirect impacts of the pandemic.

## Methods

We conducted an opportunistic before-after study, using two cross-sectional surveys conducted from February to March 2020 (survey 1), and January to March 2021 (survey 2) in Faridpur District, Bangladesh. The surveys form the baseline for the D:Clare trial, a randomised controlled trial of a community-based participatory learning and action intervention to reduce T2DM (ISRCTN 42219712) [20]. The trial was interrupted in March 2020, and resumed in January 2021 with a new baseline survey, allowing us to compare population T2DM and other NCD prevalence and risks immediately before the pandemic was declared in Bangladesh, and one year later.

### Setting

The study is based in Alfadanga upazilla (sub-district), Faridpur district, with an approximate population of 120,000. The setting was purposefully selected for the D:Clare trial as it had not

previously been exposed to the intervention, is less prone to flooding and is close to a field office [20]. Alfadanga is generally reflective of a 'typical' rural context, with an agricultural economy, and majority Bengali and Muslim population. The upazilla is divided in six unions, which we further divided into 12 clusters, and villages from all clusters were purposefully selected for inclusion on the basis of being typical of the setting. Government healthcare is provided at Union Health and Family Welfare Centres and at Community Clinics. In- and out-patient services are provided at the upazilla health complex, and tertiary care is provided at district hospitals and medical college hospitals, available in Faridpur town.

## Sampling

We followed a three-stage sampling procedure. Firstly, villages were purposefully selected from each of the 12 clusters using a 'fried egg' approach to minimise contamination between intervention and control clusters for the main D:Clare trial. We aimed for 800–1000 households in each cluster with eligible villages defined as: not sitting on a border with a neighbouring study cluster, not a major trading centre or administrative centre, and having a minimum of 50 households. The list of villages, and their estimated population sizes was derived from the 2011 Bangladesh census, and administrative maps. A sampling frame of all the households within the selected villages was then generated through our own study household census completed in November 2019. A simple random sample of households with at least one eligible adult was done, followed by the selection of a single eligible adult from sampled households, again using simple random sampling. Eligible adults were aged 30 years or older, resident in Alfadanga for at least 6-months, and not pregnant. At each survey, a new sample of households and individuals was generated, and some individuals may be sampled in both surveys by chance. The sample for the first survey was 1320 and 1584 for the second survey, according to the D:Clare trial which assumed a baseline prevalence of increased blood glucose of 40% and power to detect a 30% reduction [20].

## Data collection

Both surveys followed the same procedures. Local data collectors underwent 10 days training, including: consent, survey tools and taking anthropometric measurements, followed by one-week field piloting in a neighbouring upazilla. Data collectors worked in teams of two, with one female and one male data collector. Three field supervisors were responsible for observing and verifying data within each team at least every two days. The sampled individuals were informed of the study procedures the day before planned data collection by the study data collectors, requested to fast overnight, and asked to wear a light layer of loose clothing, which was worn for measurements. We planned two visits to each community, with the second seeking to recruit those missed in the first visit. We did not provide any monetary or non-monetary incentives for participation.

The data collectors conducted anthropometric measurements and administered a survey, adapted from the WHO Stepwise tool [21], and the 2014 Bangladesh Demographic and Health Survey [22]. This included questions on: socio-demographics; lifestyle and behavioural risk factors (diet, exercise, tobacco use); T2DM diagnosis, treatment, complications and care-seeking; hypertension and obesity diagnosis and other NCD care-seeking. Given the focus of the D:Clare Trial, more in-depth questions were asked about T2DM compared to other pre-existing NCDs. Depression and anxiety were assessed using the GAD-7 and PHQ-9 mental health screening tools amongst all participants [23, 24], the EQ-5D visual analogue scale used for self-rated health [25]. In survey 2 only, we asked about changes in recent care-seeking and avoidance of care.

For anthropometric data collection, temporary testing locations in a convenient central location in the village were set-up. Anthropometric measures included: height, weight, waist and hip circumference, blood pressure and fasting and 2-hour post-glucose load blood glucose. Blood pressure was measured twice, with a 5-minute interval between measurements using the Omron HBP 1100 Professional Blood Pressure Monitor (Kyoto, Japan). Blood glucose was measured from whole capillary blood from the finger with the OneTouch Verio Flex Glucometer (Lifescan IP Holdings, LLC), which was calibrated before data collection. Participants received a 75g glucose load dissolved in 300ml of water, and had a second blood glucose sample taken 2-hours later. People who self-reported diabetes were exempt from fasting and taking the glucose load, and instead had a single random blood glucose test. All data was collected using ODK Collect and was uploaded to a central server on a weekly basis for routine cleaning and checks.

## Analysis

Definitions of the categorical and binary indicators for impaired glucose tolerance and T2DM, hypertension, obesity and abdominal obesity, anxiety and depression are presented in **Table 1**. The primary analysis described these outcomes as proportions and 95% confidence intervals and compared between the surveys using chi2 tests. We conducted a multivariable logistic regression to estimate the association of survey round (exposure) on the NCD outcomes,

**Table 1. Definitions for anthropometric measures.**

| Variable | | Definition |
|---|---|---|
| Diabetes status [40] | Normal | Fasting plasma glucose <6.1mmol/l |
| | Impaired Fasting Glucose | Fasting plasma glucose ≥6.1mmol/l to <7.0mmol/l |
| | | *and* |
| | | Two-hour post glucose load blood glucose of <7.8mmol/l |
| | Impaired Glucose Tolerance | Fasting plasma glucose <7.0mmol/l |
| | | *and* |
| | | Two-hour post glucose load blood glucose of ≥7.8mmol/l to <11.1mmol/l |
| | Type 2 Diabetes Mellitus | Fasting plasma glucose ≥7.0 mmol/l |
| | | *Or* |
| | | Two-hour post glucose load blood glucose of ≥11.1mmol/l |
| | | *Or* |
| | | Self-reported diagnosis of diabetes by a healthcare provider |
| Hypertension* | | Systolic blood pressure ≥ 140 mmHg |
| | | *Or* |
| | | Diastolic blood pressure ≥ 90 mmHg |
| | | *Or* |
| | | Self-reported diagnosis of hypertension by a healthcare provider |
| Obesity | Underweight | Body mass index <18.5 |
| | Normal | Body mass index ≥18.5 to <23 |
| | Overweight | Body mass index ≥23 to <28 |
| | Obese | Body mass index ≥28 |
| Abdominal obesity | | Waist to hip ratio ≥0.85 for woman and ≥0.9 for men |
| Anxiety [23] | | GAD7 score ≥10 |
| Depression [41] | | PHQ9 score ≥10 |

*The average of the two blood pressure measurements was used

adjusted for socio-demographic factors (gender, age, socio-economic tertile, education, occupation and marital status). We used a complete case analysis due to low numbers of missing data. Secondary analysis of continuous outcomes of fasting glucose (mmol/l), systolic and diastolic blood pressure (mmHg), body mass index (BMI), waist-to-hip ratio, self-rated health (scale from 0–100), PHQ-9 and GAD-7 summarised the mean and standard deviation, and compared values in survey 1 and 2 using linear regression.

Descriptions of self-reported food consumption, exercise, NCD diagnosis awareness, T2DM treatment and complications, care-seeking and associated costs are presented. Exploratory sub-group analyses were done by gender, age and socio-economic tertiles, to check if the direction and magnitude of change were broadly similar across groups. Data was weighted to account for the sampling method (Appendix A in S1 File). Analyses were conducted using Stata SE14.

## Ethics

Written informed consent was obtained from all participants before data collection, or a thumb print for those unable to write. Ethical approvals were given by the University College London Research Ethics Committee (ref: 4199/007) and the Ethical Review Committee of the Diabetic Association of Bangladesh (ref: BADAS-ERC/E/19/00276). Additional information regarding the ethical, cultural, and scientific considerations specific to inclusivity in global research is included in (S1 Questionnaire).

## Results

### Study participants

We recruited 950 (72.0%) and 1392 (87.9%) participants in the first and second surveys (**Fig 1**). In both surveys, non-responders were more commonly male (55.2% and 60.4%). Socio-demographic characteristics were similar across surveys, except for the proportion of currently married respondents (92.6% in survey 1 versus 86.8% in survey 2)–**Table 2**.

### Diabetes

The prevalence of T2DM and mean population fasting glucose was similar between the two surveys, and no significant differences were seen when adjusted for socio-demographic factor or explored by sub-group (**Tables 3 and 4, Appendices B to D in S1 File**). Overall half of participants with T2DM based on the glucose testing were aware of their status (48.6%). While more diabetic participants were aware of their diagnosis in the first survey (53.5%; 95% CI: 47.8, 59.2), this was not statistically different from survey two (45.3%; 95% CI: 37.4, 53.5; p-value = 0.102). Additionally, there was no difference in time since diagnosis, with 14.9% (95% CI: 8.1, 25.7) in survey 1 and 13.1% (95% CI: 6.4, 24.8; p-value = 0.772) in survey 2 diagnosed in the prior 12-month period.

Of the whole population 196 people self-reported diabetes. They reported no differences in their history of diabetes-related complications (76.3% (95% CI: 66.0, 84.2) versus 72.4% (95% CI: 54.6, 85.1); p-value = 0.656) or hospitalisation in the prior year (4.1% (95% CI: 1.0, 16.0) versus 3.3% (95% CI: 1.2, 9.2); p-value = 0.807). In terms of self-care, blood glucose testing at least monthly decreased from 46.0% (95% CI: 33.9, 58.5) to 30.5% (95% CI: 23.4, 38.7; p-value<0.001), but the proportion of people with diabetes taking any medication increased non-significantly from 68.9% (95% CI: 54.2, 80.6) to 73.5% (95% CI: 63.8, 81.3; p-value = 0.552). The average amount spent in the prior 30-days on care-seeking for diabetes was significantly lower in survey two (mean: 888 Bangladeshi Taka (BDT); 95% CI: 737, 1039)

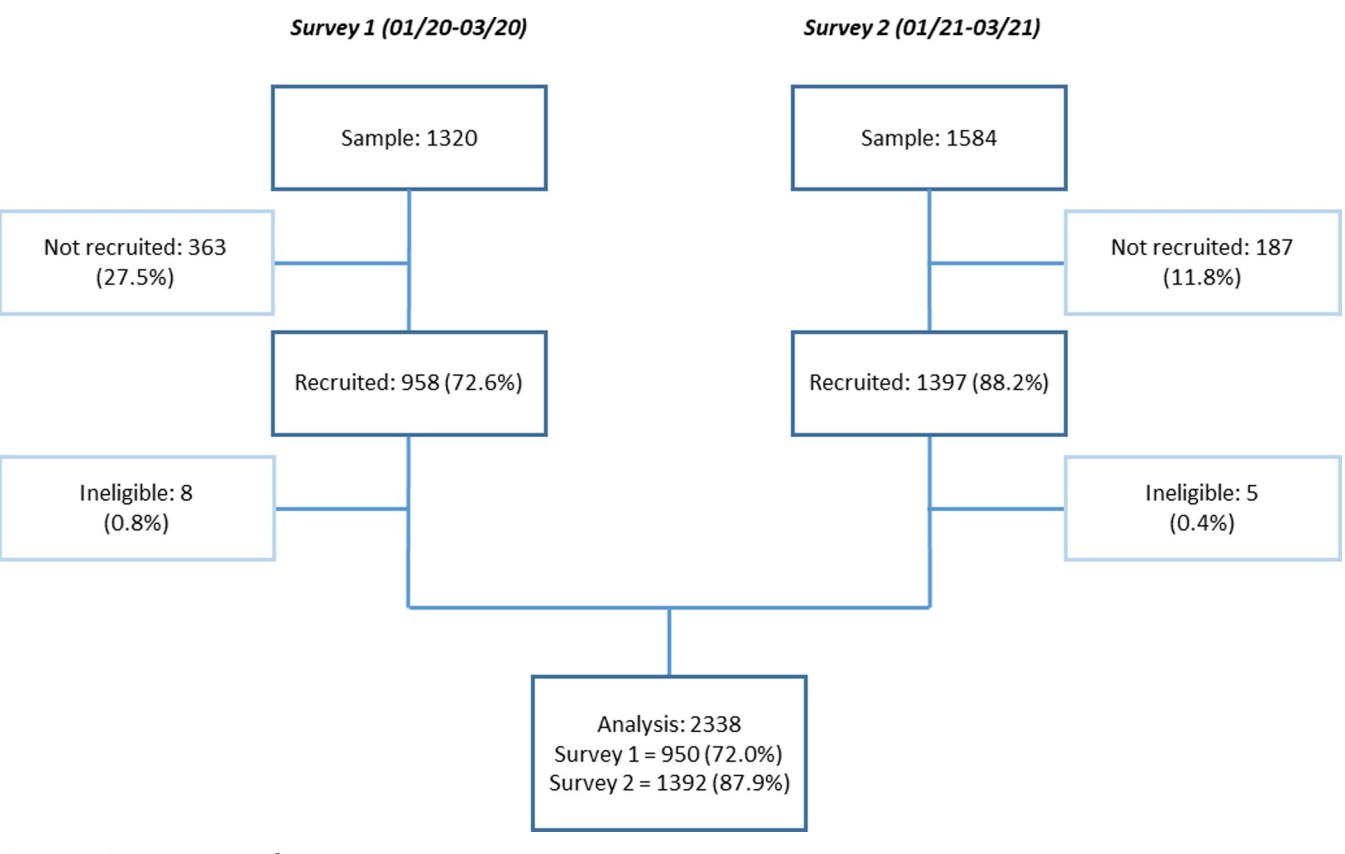

**Fig 1. Participant recruitment diagram.**

compared with survey one (mean: 1565 BDT; 95% CI: 938, 2191; p-value = 0.040). In the second survey, 20.0% (n = 22/117) of self-reported people with diabetes reported delaying or avoiding care in the prior 6-months; specifically, 3.2% (n = 5/117) reported concerns about COVID-19 and lockdown restrictions as the reason.

## Hypertension and obesity

The percentage of people with hypertension significantly increased from 34.5% in survey 1 to 41.5% in survey 2 (**Table 3**), and this association was statistically significant when adjusted for socio-demographic factors in the multivariable regression (aOR: 1.31; 95% CI: 1.04, 1.66 – **Table 4**). In the sub-group analysis, the increase in hypertension was more pronounced for men (+9.9%, p-value = 0.012), than women (+4.8%, p-value = 0.148)–**Fig 2**, **Appendix B in S1 File**, but was a similar percentage point increase amongst age and wealth sub-groups (**Appendices C and D in S1 File**). This increase was also reflected in the mean population systolic and diastolic measures. Amongst the participants who were hypertensive according to the blood pressure measurements, 48.2% had been previously diagnosed, and this was lower in survey two (43.2%; 95% CI: 34.0, 53.0) compared to survey one (57.3%; 95% CI: 49.7, 64.6; p-value = 0.027).

Overall there were no significant changes in either BMI or abdominal obesity categories. Women, the wealthiest tertile and younger respondents had higher rates of both measures than men or older persons, respectively. Of those with a BMI>23, 17.5% had been previously told that they were overweight by a healthcare provider, with no change across survey one (17.8%; 95% CI: 10.0, 29.7) and survey two (17.3%; 95% CI: 10.6, 27.0; p-value = 0.928).

**Table 2. Demographic characteristics of participants in both surveys.**

| | | Survey 1 (N = 948)* | | Survey 2 (N = 1392) | |
|---|---|---|---|---|---|
| Age group | 30–39 | 298 | 29.7% | 414 | 26.6% |
| | 40–49 | 231 | 23.4% | 352 | 24.8% |
| | 50–59 | 180 | 19.5% | 244 | 18.2% |
| | 60–69 | 157 | 18.1% | 243 | 18.8% |
| | >70 | 82 | 9.3% | 139 | 11.6% |
| Gender | Female | 576 | 58.1% | 876 | 60.3% |
| | Male | 372 | 41.9% | 516 | 39.7% |
| Married | No | 76 | 7.4% | 210 | 13.2% |
| | Yes | 872 | 92.6% | 1182 | 86.8% |
| Wealth quintile | Most poor | 195 | 19.2% | 274 | 18.6% |
| | Very poor | 218 | 22.6% | 262 | 19.3% |
| | Poor | 174 | 19.3% | 281 | 20.6% |
| | Less poor | 183 | 18.6% | 285 | 20.3% |
| | Least poor | 177 | 20.3% | 290 | 21.3% |
| Education | None | 590 | 63.0% | 826 | 61.0% |
| | Primary | 261 | 26.7% | 423 | 29.0% |
| | Secondary / further | 97 | 10.2% | 143 | 10.1% |
| Occupation | Not working | 611 | 63.0% | 925 | 66.0% |
| | Manual labour | 303 | 33.8% | 407 | 29.5% |
| | Professional / skilled | 32 | 3.3% | 60 | 4.5% |
| Muslim | No | 45 | 4.9% | 44 | 3.7% |
| | Yes | 903 | 95.1% | 1348 | 96.3% |

*Two participants did not complete the survey

The average amount spent in the prior 30-days on care for non-T2DM NCDs was again lower in survey two (799 BDT; 95% CI: 640, 959) compared to survey one (1966 BDT; 95% CI: 724, 3208; p-value = 0.066). Amongst those with another self-reported NCD diagnosis, 5.3% (n = 19/356) reported delaying or avoiding care in the prior 6-months due to concerns about COVID-19 or lockdown.

## Depression, anxiety and self-rated health

Across all three measures of self-reported health and mental health, there was a significant improvement between survey one and two (**Fig 3**). For self-rated health, assessed on a scale of 0–100, we observed a significant 10-point increase (71.3 vs 81.2, p-value = 0.005 –**Table 3**), and this was consistent across all sub-groups; the largest absolute increase was +14.6 points in the lowest wealth tertile group (**Appendix D in S1 File**). Depression was twice as high in the first survey, and generalised anxiety was 4-times higher–**Table 3**. The decreases in depression and anxiety were larger for women, who had lower self-rated health, and considerably higher rates of depression and anxiety in survey one than men (**Appendix B in S1 File**). The odds of anxiety were 83% lower in survey 2 (aOR: 0.17; 95% CI: 0.07, 0.41), and 72% lower for depression (aOR: 0.28; 95% CI: 0.10, 0.82)–**Table 4**.

## Diet and exercise

Diet and exercise variables are summarised in **Table 5**. Overall no significant difference in the hours of physical activity was seen between surveys. However, there was a significant

**Table 3. Prevalence of non-communicable conditions in survey 1 and survey 2.**

| Condition | | Survey 1 (n = 948)[*] | | Survey 2 (n = 1392) | | p-value |
|---|---|---|---|---|---|---|
| | | N | % (95% CI) | N | % (95% CI) | |
| Diabetes | Normal | 550 | 57.8% (52.8, 62.7) | 811 | 57.9% (53.0, 62.8) | 0.437 |
| | IFG | 41 | 3.8% (2.4, 6.1) | 64 | 5.2% (4.0, 6.6) | |
| | IGT | 203 | 21.6% (18.6, 24.9) | 262 | 19.0% (16.0, 22.5) | |
| | T2DM | 153 | 16.8% (14.8, 18.9) | 250 | 17.9% (15.0, 21.2) | |
| Hypertension | | 292 | 34.5% (30.7, 38.5) | 537 | 41.5% (38.2, 45.0) | 0.011 |
| BMI | Underweight | 114 | 12.4% (10.3, 14.8) | 162 | 12.9% (10.9, 15.2) | 0.936 |
| | Normal | 391 | 42.0% (39.4, 44.7) | 562 | 40.8% (38.3, 43.4) | |
| | Overweight | 346 | 35.9% (32.8, 39.1) | 521 | 36.2% (33.0, 39.5) | |
| | Obese | 94 | 9.7% (7.4, 12.6) | 146 | 10.1% (7.8, 13.0) | |
| Abdominal obesity | | 600 | 63.0% (58.1, 67.7) | 939 | 66.7% (60.5, 72.4) | 0.327 |
| Depression | | 153 | 15.2% (8.4, 26.1) | 84 | 6.0% (2.7, 12.6) | 0.048 |
| Anxiety | | 177 | 17.9% (11.3, 27.3) | 60 | 4.0% (2.0, 7.6) | <0.001 |
| | | Mean | (95% CI) | Mean | (95% CI) | p-value |
| Fasting blood glucose | | 5.86 | (5.64, 6.08) | 5.86 | (5.64, 6.09) | 0.969 |
| Systolic blood pressure | | 127.41 | (125.27, 129.55) | 134.25 | (132.18, 136.32) | <0.001 |
| Diastolic blood pressure | | 72.97 | (71.46, 74.48) | 77.42 | (76.05, 78.80) | <0.001 |
| BMI | | 22.85 | (22.56, 23.14) | 22.88 | (22.53, 23.23) | 0.871 |
| Waist to hip ratio | | 0.89 | (0.89, 0.90) | 0.89 | (0.88, 0.90) | 0.891 |
| Self-rated health | | 71.34 | (66.82, 75.86) | 81.17 | (76.33, 86.01) | 0.005 |
| PHQ-9 score | | 3.27 | (2.14, 4.40) | 1.85 | (1.03, 2.66) | 0.045 |
| GAD-7 score | | 5.73 | (4.75, 6.71) | 3.57 | (2.88, 4.25) | 0.001 |

Data presented are weighted for the sampling methodology.

[*]Two participants did not complete the anthropometric measurements

decrease amongst men, from 14.3 hours (95% CI: 9.8, 18.8) to 9.2 hours (95% CI: 7.3, 11.1; p-value = 0.041). Dietary diversity and rice consumption was lower in survey two, while household consumption of oil and added sugar increased, and added salt and unhealthy snacks showed no change. These patterns were consistent in gender, age and wealth sub-groups.

**Table 4. Multivariable logistic regressions of NCD outcomes and survey exposure, adjusted for socio-demographic factors.**

| Outcome | aOR[*] | 95% CI | p-value |
|---|---|---|---|
| Abnormal glucose[**] | 0.97 | 0.72, 1.31 | 0.834 |
| Hypertension | 1.31 | 1.04, 1.66 | 0.026 |
| Obesity | 1.04 | 0.71, 1.52 | 0.851 |
| Abdominal obesity | 1.16 | 0.82, 1.64 | 0.383 |
| Anxiety | 0.17 | 0.07, 0.41 | <0.001 |
| Depression | 0.28 | 0.10, 0.82 | 0.022 |

Data presented are weighted for the sampling methodology.

[*]All models were adjusted for: age, gender, socio-economic tertile, education, occupation and marital status.

[**]Combined outcome of impaired fasting glucose (IFG), impaired glucose tolerance (IGT), and type 2 diabetes (see Table 1 for definitions).

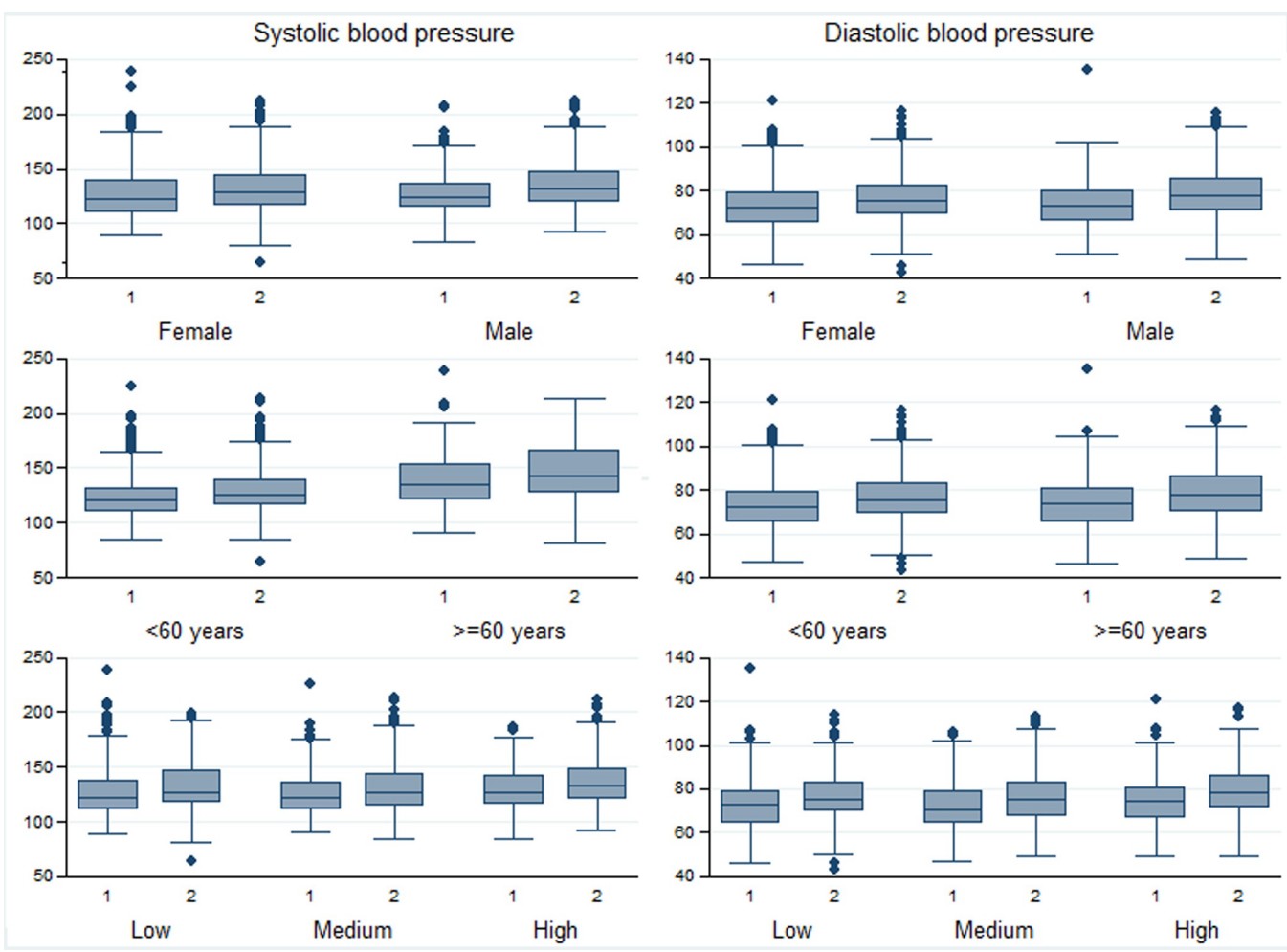

**Fig 2. Systolic and diastolic blood pressure, by gender, age and wealth sub-groups.**

## Discussion

In this opportunistic comparison of pre-COVID-19 and post-wave 1 lockdown population-level prevalence of NCDs and NCD risk factors in a rural Bangladesh setting, we observed a worsening in blood pressure. We did not see increases in blood glucose, diabetes, obesity or under-weight, and surprisingly, improvements across all self-reported indicators of health and mental health were observed one year on from the first COVID-19 cases in Bangladesh. Some differences in diet were seen, and less money spent on NCD care suggest negative economic impacts of the pandemic led to changes in self-care.

The increase in hypertension, and a decline in those who were aware of their diagnosis, from 57% to 43%, suggests the pandemic may have worsened the diagnostic gap. This increase in hypertension was also more pronounced in men than women, and may be explained by the reduction in exercise, in combination with the decline in dietary quality (i.e. increased sugar and oil consumption and decreased dietary diversity). However, self-reported salt consumption did not change, despite being one of the best established risks for developing hypertension [26]. The absolute increase in systolic (6.84 mmHg) and diastolic (4.45 mmHg) population measures is a considerable cause for concern given strong associations with cardiovascular

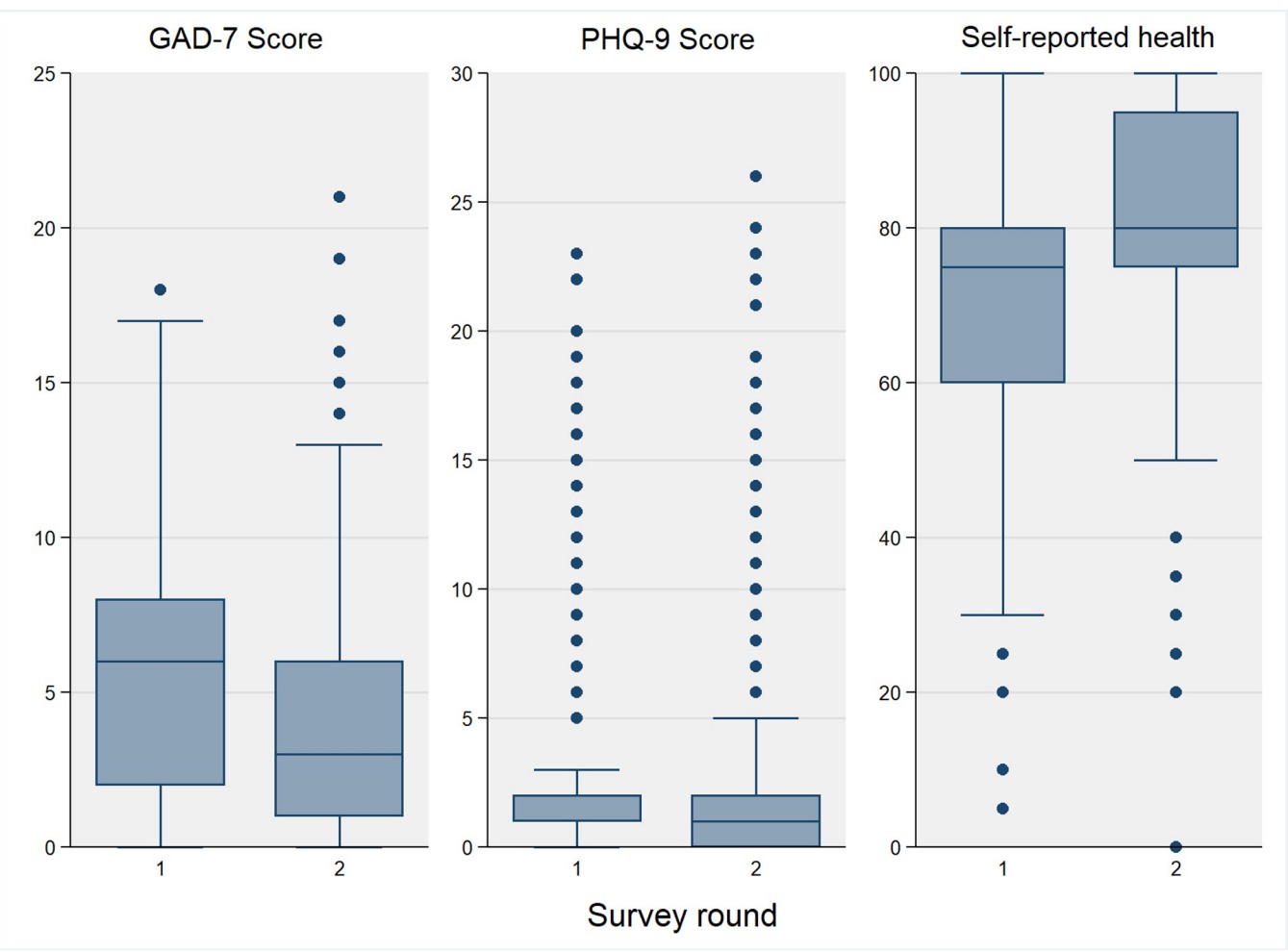

**Fig 3. Change in self-reported health, GAD-7 an PHQ-9 between survey 1 and 2.**

morbidity and mortality [27]. Therefore, it should be a priority to diagnose and treat missed hypertension cases, together with population-level risk reduction strategies.

Our finding of 2–4 times lower prevalence of depression and anxiety was unexpected and counters the common narrative around mental health and COVID-19. Additionally the 7–18 point increases in self-rated health in the second survey, given objective increases in hypertension (suggesting poorer health), also seems contradictory. Globally, the COVID-19 pandemic has been widely reported as worsening mental health, with an estimated 36.1% increase in depression and 35.1% increase in anxiety cases in the South Asian region [28]. Indeed, Bangladesh was estimated to have had one of the largest increases in poor mental health, but notably no primary data from the South Asian region was used in this modelled estimate. Studies from Bangladesh conducted during the lockdown period have reported high rates of depression in the general population (27.8%-34.1%) [14, 29–31]. However, two used online convenience sampling, making direct comparisons with our rural representative adult population challenging. The other key difference between our study and this literature is that our second survey occurred after restrictions had been lifted and before the second wave.

Appropriate published evidence to compare our findings with has been hard to find, and therefore we can only speculate on the reasons for improved mental health. One hypothesis is

**Table 5. Active work, exercise and dietary consumption in survey 1 and 2.**

| Exercise (hrs per week) | | Survey 1 | | | Survey 2 | | | p-value |
|---|---|---|---|---|---|---|---|---|
| | | Mean | 95% CI | | Mean | 95% CI | | |
| Active work | | 9.62 | (6.16, | 13.07) | 8.17 | (6.44, | 9.90) | 0.447 |
| Sport | | 0.64 | (0.40, | 0.88) | 0.48 | (0.19, | 0.78) | 0.407 |
| Total active (work + sport) | | 10.26 | (6.67, | 13.84) | 8.65 | (6.93, | 10.38) | 0.413 |
| Total sitting (daily) | | 2.03 | (1.65, | 2.42) | 2.20 | (1.92, | 2.47) | 0.480 |
| **Diet** | | **Mean** | **95% CI** | | **Mean** | **95% CI** | | |
| Dietary diversity* | | 7.89 | (7.46, | 8.31) | 6.90 | (6.59, | 7.20) | 0.001 |
| Rice (grams/day) | | 1078.5 | (984.0, | 1173.1) | 785.7 | (737.1, | 834.2) | <0.001 |
| Bread (pieces/day) | | 0.59 | (0.45, | 0.74) | 0.60 | (0.47, | 0.73) | 0.983 |
| Oil (litres/mth) | | 4.45 | (4.27, | 4.63) | 4.78 | (4.62, | 4.95) | 0.010 |
| | | % | 95% CI | | % | 95% CI | | |
| Snack consumption** | | 56.3% | (49.9, | 62.5) | 56.8% | (50.7, | 62.7) | 0.911 |
| Added salt | Every meal | 43.7% | (38.3, | 49.3) | 45.1% | (41.1, | 49.2) | 0.986 |
| | Most meals | 13.1% | (9.2, | 18.3) | 12.8% | (8.9, | 18.0) | |
| | Occasionally | 20.8% | (16.5, | 26.0) | 20.1% | (15.3, | 26.0) | |
| | Never | 22.4% | (17.2, | 28.6) | 22.0% | (17.2, | 27.7) | |
| Added sugar | None | 70.7% | (62.7, | 77.6) | 60.0% | (55.8, | 63.9) | <0.001 |
| | 1 teaspoon | 12.6% | (9.4, | 16.6) | 12.0% | (7.6, | 18.4) | |
| | 2 teaspoons | 3.6% | (2.4, | 5.4) | 17.2% | (13.6, | 21.4) | |
| | 3 or more teaspoons | 5.8% | (3.9, | 8.5) | 9.6% | (6.7, | 13.4) | |
| | Not sure | 7.4% | (2.1, | 22.6) | 1.4% | (0.6, | 2.8) | |

Data presented are weighted for the sampling methodology.

*Number of the following food items eaten the day before the survey: rice, bread, tubers, pulses, nuts, milk and dairy products, offal, meat and poultry, fish, eggs, ripe fruits, other fruits, leafy vegetables, root vegetables, salad vegetables, sugar, and other foods (total = 17 items).

**Consumption of 'unhealthy' snacks the day before the survey, including: puri, samosa, crisps, biscuits, chocolate or sweets, sugary drink, puffed rice, chira, payesh, noodles.

that a sense of relief that restrictions had been lifted, case numbers were low and study participants had survived meant participants reported more positively. This may especially hold true if communities trusted that restrictions had protected them, and therefore the lifting of lockdown reflects a safer situation. An online survey from New Zealand reported post-lockdown pride in coping and appreciation of family [32], and a UK study observed living in rural areas and with others were protective against lockdown loneliness [33]. Therefore, the return of family members from international and urban areas to their rural family homes during periods of restriction [34], may also have been a key contributing factor. A qualitative contextual understanding of this finding is needed.

The only methodological difference between the two surveys was five of the 12 data collectors being replaced, and it is possible that interviewers influenced participant's respondents through the way they administered the questions. Considerable efforts were made during training to standardise the interview process, and both the GAD-7 and PHQ-9 have been used and validated for the Bangladeshi context [35, 36]. Given the results were consistent across sub-groups and all three measures, this suggests it is a genuine change. Our data should highlight the need for on-going assessments of mental health impacts, to understand care and support needs over the course of this complex pandemic situation.

We observed two possible financial coping strategies—reduced food consumption and reduced care-seeking. While respondents in the second survey did not commonly report

delaying care-seeking due to COVID-19 concerns, they spent less money on care and people living with T2DM reduced their glucose testing. In this context, informal observations suggest the price of medications can be lower than going for routine blood glucose testing, and so as a financial coping mechanism it is plausible that people known to have diabetes shifted their management priorities. This is reflected in a study in which people with diabetes reported that they were more concerned about not being able to test (26%) due to COVID-19 than access to medication (18%), and a quarter felt their quality of care had declined [12]. Other studies from Bangladesh and India have also reported reduced care access [11, 13, 37]. However multiple barriers to accessing diabetes services were present before the pandemic, including cost [38, 39]. This short term shift in management, and challenges in care access, did not appear to lead to catastrophic health outcomes for known diabetics. However, this needs to be carefully monitored for possible longer term consequences.

While our study is strengthened by the large random population sample, and objective anthropometric measures, we had three key limitations. Firstly, the lower response rate in the first survey. We stopped field activities early due to the COVID-19 pandemic, and therefore had a higher proportion of sampled participants who were not recruited. The age and gender of non-responders was similar between survey 1 and 2, but the non-responders may differ in other socio-demographic factors which made them hard to recruit on our first visit their village. Secondly, we relied on self-reported measures of diet, exercise and care-seeking, depression and anxiety were classified using screening tools instead of clinical diagnosis and we considered self-reported clinical diagnoses of diabetes and hypertension as valid. These variables are therefore subject to recall and social-desirability biases and non-differential misclassification of outcomes; however, given our methodology was the same for both surveys we could expect these biases to be consistent. Third, the change in NCD prevalence may reflect regression to the mean, with the sample in survey 1 representing values further from the true population mean.

Amongst this representative rural adult population from Faridpur district, Bangladesh, it was apparent that several months after COVID-19 lockdown measures had been lifted, both negative and positive health changes were still present. Most notably, between the 12-month period from early 2020 to 2021 there had been a large population increase in systolic and diastolic blood pressure. The potential public health implications of this are substantial and need to be a priority for both preventive risk reduction and diagnosis and treatment of existing morbidities. However, on a positive note, consistent improvements in mental health suggest that the widely document negative impacts of lockdowns may not persist once restrictions are lifted. Further work into understanding this phenomenon is needed to support effective recovery.

## Supporting information

**S1 File.** Appendix A. Summary of survey weighting. Appendix B. Anthropometric and mental health measures, by gender. Appendix C. Anthropometric and mental health measures, by age group. Appendix D. Anthropometric and mental health measures, by wealth group.
(DOCX)

**S1 Questionnaire. Inclusivity in global research.**
(DOCX)

## Acknowledgments

We thank the members of our trial steering committee: Dr. David Beran (Chair), Prof. Graham Hitman, Prof. Sarah Hawkes, Prof. Anthony Costello, Prof. Edward Gregg, Dr. Jennifer

Thompson, Prof. Audrey Prost. We would also like to acknowledge the communities and community leaders for their engagement with the project.

## Author Contributions

**Conceptualization:** Carina King, Sanjit Kumer Shaha, Joanna Morrison, Naveed Ahmed, Abdul Kuddus, Malini Pires, Tasmin Nahar, Raduan Hossin, Hassan Haghparast-Bidgoli, Kishwar Azad, Edward Fottrell.

**Data curation:** Carina King, Sanjit Kumer Shaha, Naveed Ahmed.

**Formal analysis:** Carina King, Edward Fottrell.

**Funding acquisition:** A. K. Azad Khan, Kishwar Azad, Edward Fottrell.

**Project administration:** Sanjit Kumer Shaha, Abdul Kuddus, A. K. Azad Khan, Kishwar Azad, Edward Fottrell.

**Supervision:** Sanjit Kumer Shaha, Naveed Ahmed, Abdul Kuddus, Kishwar Azad.

**Writing – original draft:** Carina King.

**Writing – review & editing:** Sanjit Kumer Shaha, Joanna Morrison, Naveed Ahmed, Abdul Kuddus, Malini Pires, Tasmin Nahar, Raduan Hossin, Hassan Haghparast-Bidgoli, A. K. Azad Khan, Justine Davies, Kishwar Azad, Edward Fottrell.

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
