## [Decision Letter · Decision Letter 0]

18 Feb 2022

PGPH-D-21-01147

Changes in non-communicable diseases, diet and exercise in rural Bangladesh before and after the first wave of COVID-19

Dear Dr. King,

Thank you for submitting your manuscript to PLOS Global Public Health. After careful consideration, we feel that it has merit but does not fully meet PLOS Global Public Health’s publication criteria as it currently stands. Therefore, we invite you to submit a revised version of the manuscript that addresses the points raised during the review process.

We look forward to receiving your revised manuscript.

Kind regards,

Rajat Das Gupta, M.D.

Academic Editor

Journal Requirements:

2. We noticed that you used "unpublished data" in the manuscript. We do not allow these references, as the PLOS data access policy requires that all data be either published with the manuscript or made available in a publicly accessible database. Please either remove these references, or amend the supplementary material to include the referenced data.

3. Please update the completed 'Competing Interests' statement, including any COIs declared by your co-authors. If you have no competing interests to declare, please state "The authors have declared that no competing interests exist". 

4. We have noticed that you have uploaded supporting information but you have not included a list of legends.  Please add a full list of legends for all supporting information files (including figures, table and data files) after the references list. 

5. In the online submission form, you indicated that "These data are part of an on-going randomised controlled trial and will be made openly accessible once the main trial results have been published (expected early 2023). In the meantime, data will be made available upon reasonable request for research purposed only. Please contact Prof Ed Fottrell as the Principal Investigator (e.fottrell@ucl.ac.uk).". All PLOS journals now require all data underlying the findings described in their manuscript to be freely available to other researchers, either 1. In a public repository, 2. Within the manuscript itself, or 3. Uploaded as supplementary information.

6. Please amend your detailed Financial Disclosure statement. This is published with the article, therefore should be completed in full sentences and contain the exact wording you wish to be published.

ii). State the initials, alongside each funding source, of each author to receive each grant.

iii). State what role the funders took in the study. If the funders had no role in your study, please state: “The funders had no role in study design, data collection and analysis, decision to publish, or preparation of the manuscript.”

Additional Editor Comments (if provided):

1. In the hypertension definition, did the authors consider those as hypertensives, who were taking antihypertensive medications but had normal blood pressure?

2. Please add a supplementary table to describe the weighting methods.

3. In the tables, are the frequencies and percentages weighted or unweighted? Please mention that in the footnote.

Reviewers' comments:

Reviewer's Responses to Questions

**Comments to the Author**

1. Does this manuscript meet PLOS Global Public Health’s publication criteria? Is the manuscript technically sound, and do the data support the conclusions? The manuscript must describe methodologically and ethically rigorous research with conclusions that are appropriately drawn based on the data presented.

Reviewer #1: Yes

Reviewer #2: Yes

2. Has the statistical analysis been performed appropriately and rigorously?

Reviewer #1: Yes

Reviewer #2: Yes

3. Have the authors made all data underlying the findings in their manuscript fully available (please refer to the Data Availability Statement at the start of the manuscript PDF file)?

Reviewer #1: Yes

Reviewer #2: Yes

4. Is the manuscript presented in an intelligible fashion and written in standard English?

Reviewer #1: Yes

Reviewer #2: Yes

5. Review Comments to the Author

Reviewer #1: Dear Authors,

Thank you for submitting this well-written manuscript. I believe it will be a good addition to the global pool of literature.

Hence, I am happy to recommend it for publication.

Best,

Reviewer

Reviewer #2: The concept of this manuscript is good. Two issues should be taken into consideration. First of all graphical description of some analysis is more suitable. For example, if the increasing rate of hypertension or the lower prevalence of depression and anxiety could be presented via some plots that would be better. The second one is about the improvement in mental health on the second survey. This finding contradicts the typical scenario of the whole world at that time. The reason provided behind it does not seem to be so strong. It needs to be reviewed again.

6. PLOS authors have the option to publish the peer review history of their article (what does this mean?). If published, this will include your full peer review and any attached files.

**Do you want your identity to be public for this peer review?** For information about this choice, including consent withdrawal, please see our Privacy Policy.

Reviewer #1: No

Reviewer #2: No

---

## [Decision Letter · Decision Letter 1]

1 Aug 2022

PGPH-D-21-01147R1

Changes in non-communicable diseases, diet and exercise in rural Bangladesh before and after the first wave of COVID-19

Dear Dr. King,

Thank you for submitting your manuscript to PLOS Global Public Health. After careful consideration, we feel that it has merit but does not fully meet PLOS Global Public Health’s publication criteria as it currently stands. Therefore, we invite you to submit a revised version of the manuscript that addresses the points raised during the review process.

We look forward to receiving your revised manuscript.

Kind regards,

Rajat Das Gupta, M.D.

Academic Editor

Journal Requirements:

Reviewers' comments:

Reviewer's Responses to Questions

**Comments to the Author**

1. If the authors have adequately addressed your comments raised in a previous round of review and you feel that this manuscript is now acceptable for publication, you may indicate that here to bypass the “Comments to the Author” section, enter your conflict of interest statement in the “Confidential to Editor” section, and submit your "Accept" recommendation.

Reviewer #1: All comments have been addressed

Reviewer #2: (No Response)

Reviewer #3: All comments have been addressed

Reviewer #4: (No Response)

2. Does this manuscript meet PLOS Global Public Health’s publication criteria? Is the manuscript technically sound, and do the data support the conclusions? The manuscript must describe methodologically and ethically rigorous research with conclusions that are appropriately drawn based on the data presented.

Reviewer #1: Yes

Reviewer #2: Yes

Reviewer #3: Yes

Reviewer #4: Yes

3. Has the statistical analysis been performed appropriately and rigorously?

Reviewer #1: Yes

Reviewer #2: Yes

Reviewer #3: Yes

Reviewer #4: Yes

4. Have the authors made all data underlying the findings in their manuscript fully available (please refer to the Data Availability Statement at the start of the manuscript PDF file)?

Reviewer #1: Yes

Reviewer #2: Yes

Reviewer #3: Yes

Reviewer #4: Yes

5. Is the manuscript presented in an intelligible fashion and written in standard English?

Reviewer #1: Yes

Reviewer #2: Yes

Reviewer #3: Yes

Reviewer #4: Yes

6. Review Comments to the Author

Reviewer #1: Dear Authors,

Thank you for submitting this well-revised version of the manuscript.

I am happy to recommend it for publication.

Best,

Reviewer

Reviewer #2: (No Response)

Reviewer #3: I appreciate the feedback on the points made in this article and with these updates I am happy to recommend it for publication.

Reviewer #4: Thank you for conducting this study and submitting it to PLOS Global Health! Overall. This study certainly is very rigorous and contains rich data, insights, and analysis. However, I have a few minor comments below addressing which, I believe, would further enrich the study.

1. Apparently. You have conducted your study in a Upazilla/sub-district of Faridpur district. Given there are 492 Upazillas in total, I wonder if the results from one Upazilla could be generalized to the entire rural Bangladesh. Furthermore, I wonder if this specific sub-district was chosen due to convenience of data collection or due to any other decision criteria. It would be great if you could present your case as to why you believe the results are generalizable – or, maybe change the title of the study accordingly.

2. Line 97-99: It would be easier for the readers and reviewers to understand if you could explain the rationale behind setting up those criteria e.g. eligible villages 98 were those which do not sit on a border with a neighbouring study cluster etc.

3. Line 117-120: Was any form of monetary/non-monetary remuneration provided to the participants?

4. Line 141-152: Did you have any missing data? If so, would you please describe how you had dealt with missing data in your analysis?

5. The inclusion of self-reported diagnosis of hypertension by a healthcare provider as one of the measures of hypertension yields the possibility of non-differential misclassification bias and recall bias. It could be useful to acknowledge that in the discussion/limitation section.

7. PLOS authors have the option to publish the peer review history of their article (what does this mean?). If published, this will include your full peer review and any attached files.

**Do you want your identity to be public for this peer review?** For information about this choice, including consent withdrawal, please see our Privacy Policy.

Reviewer #1: **Yes: **Ateeb Ahmad Parray

Reviewer #2: **Yes: **Md Nasim Saba Nishat

Reviewer #3: **Yes: **Mohammad Azmain Iktidar

Reviewer #4: No

---

## [Decision Letter · Decision Letter 2]

31 Aug 2022

Changes in non-communicable diseases, diet and exercise in a rural Bangladesh setting before and after the first wave of COVID-19

PGPH-D-21-01147R2

Dear Dr. King,

We are pleased to inform you that your manuscript 'Changes in non-communicable diseases, diet and exercise in a rural Bangladesh setting before and after the first wave of COVID-19' has been provisionally accepted for publication in PLOS Global Public Health.

Best regards,

Rajat Das Gupta, M.D.

Academic Editor

Reviewer Comments (if any, and for reference):

Reviewer's Responses to Questions

**Comments to the Author**

1. If the authors have adequately addressed your comments raised in a previous round of review and you feel that this manuscript is now acceptable for publication, you may indicate that here to bypass the “Comments to the Author” section, enter your conflict of interest statement in the “Confidential to Editor” section, and submit your "Accept" recommendation.

Reviewer #4: All comments have been addressed

2. Does this manuscript meet PLOS Global Public Health’s publication criteria? Is the manuscript technically sound, and do the data support the conclusions? The manuscript must describe methodologically and ethically rigorous research with conclusions that are appropriately drawn based on the data presented.

Reviewer #4: Yes

3. Has the statistical analysis been performed appropriately and rigorously?

Reviewer #4: Yes

4. Have the authors made all data underlying the findings in their manuscript fully available (please refer to the Data Availability Statement at the start of the manuscript PDF file)?

Reviewer #4: Yes

5. Is the manuscript presented in an intelligible fashion and written in standard English?

Reviewer #4: Yes

6. Review Comments to the Author

Reviewer #4: (No Response)

7. PLOS authors have the option to publish the peer review history of their article (what does this mean?). If published, this will include your full peer review and any attached files.

**Do you want your identity to be public for this peer review?** For information about this choice, including consent withdrawal, please see our Privacy Policy.

Reviewer #4: No
